# Staphylococcal Enterotoxin C—An Update on SEC Variants, Their Structure and Properties, and Their Role in Foodborne Intoxications

**DOI:** 10.3390/toxins12090584

**Published:** 2020-09-10

**Authors:** Danai Etter, Jenny Schelin, Markus Schuppler, Sophia Johler

**Affiliations:** 1Institute for Food Safety and Hygiene, Vetsuisse Faculty, University of Zürich, 8057 Zürich, Switzerland; sophia.johler@uzh.ch; 2Laboratory of Food Microbiology, Institute of Food, Nutrition and Health, ETH Zürich, 8092 Zürich, Switzerland; markus.schuppler@hest.ethz.ch; 3Division of Applied Microbiology, Department of Chemistry, Lund University, 22100 Lund, Sweden; jenny.schelin@tmb.lth.se

**Keywords:** staphylococcal food poisoning, SEC, superantigen, emesis, host specificity

## Abstract

Staphylococcal enterotoxins are the most common cause of foodborne intoxications (staphylococcal food poisoning) and cause a wide range of diseases. With at least six variants staphylococcal enterotoxin C (SEC) stands out as particularly diverse amongst the 25 known staphylococcal enterotoxins. Some variants present unique and even host-specific features. Here, we review the role of SEC in human and animal health with a particular focus on its role as a causative agent for foodborne intoxications. We highlight structural features unique to SEC and its variants, particularly, the emetic and superantigen activity, as well as the roles of SEC in mastitis and in dairy products. Information about the genetic organization as well as regulatory mechanisms including the accessory gene regulator and food-related stressors are provided.

## 1. Introduction

Foodborne illnesses are one of the world’s leading health issues. They are estimated to cause 420,000 deaths every year and generate health costs and economic losses in the range of 110 billion USD worldwide [1]. One of the most common causative agents for food intoxications are staphylococcal enterotoxins (SEs). These exotoxins are preformed by *Staphylococcus aureus* in food and cause intoxication upon ingestion. In the EU alone, 114 cases of foodborne outbreaks were attributed to these heat-stable SEs in 2018 [2]. Symptoms of staphylococcal food poisoning (SFP) include violent vomiting, diarrhea, fever, and unspecific symptoms like headache and nausea. Due to the generally-quick recovery of patients, the number of cases is likely being underestimated [3,4].

SEs are produced by *S. aureus* during growth, alongside a number of other virulence factors. The organism can not only cause food intoxications but also various infectious diseases including toxic shock syndrome (TSS) [5,6,7]. Staphylococcal virulence factors include exopolysaccharides, surface-associated protein adhesins, immune modulators, and exoproteins including a variety of toxins [8]. Staphylococcal virulence factors [9,10] and their role in both infectious diseases [6,11] and food intoxications [12,13] have been previously reviewed. The most relevant secreted toxins are SEs, hemolysins, leukotoxins, exfoliative toxins, and toxic shock syndrome toxin (TSST-1) [14]. So far, 25 SEs (SEA–SE*l*Z) have been described, excluding variants and TSST-1 (formerly SEF), but new types are frequently discovered [12]. Non-emetic toxins or not-yet-tested ones are referred to as SE-like (SE*l*) [15]. These peptides are all classified as pyrogenic toxin superantigens (SAgs) and have the ability to mobilize large proportions of T-cells (20–30%) [16,17,18]. The first enterotoxin SEA was isolated in 1959 [19,20], shortly followed by SEB and SEC [21,22]. All newly-discovered enterotoxins have subsequently been named alphabetically. The genetic location of SEs differs greatly between toxins and can even vary within the same type of SE (Table 1). SEA–SEE are considered "classical enterotoxins", while SEG–SE*l*Z are termed "new enterotoxins". Historically, classical toxins were demonstrated to be emetically active in rhesus monkey feeding assays, while new toxins were either not emetic or had not been tested in monkeys [12,23]. However, this distinction has become obsolete since many new toxins were later recognized as emetically active in rhesus monkey feeding assays [24], and new toxins were found as causative agents for SFP [25,26]. Additionally, smaller mammals such as ferrets or the house musk shrew were proposed as models to simplify testing [27,28]. So far, commercial antibodies can only be obtained for classical SEs.

Originally, SEs were differentiated by immunological methods, whereas today it is recommended to identify the relationship of new toxins via sequence homology (where >90% sequence homology equals a toxin variant and <90% homology determines a new toxin) [15]. SEC takes on a special role in this terminology, since several variants, often even host-specific ones, have been described [29,30]. In particular, the ruminant-adapted toxin variants SEC_bovine_ and SEC_ovine_ show the exceptional ability of *S. aureus* to adapt to different niches. SEC is by far the most-frequently-isolated toxin in animals suffering from mastitis [31,32,33,34]. Additionally, SEC alone was shown to cause inflammation, proinflammatory cytokine production, and tissue damage in mammary glands. SEC might therefore play an important role in the development of mastitis associated with *S. aureus* infection. [31]. There are some reports of SEC also being involved in human post-partum mastitis. [35]. Contaminated ruminant milk can also provide an entry-point for *S. aureus* or its toxins into the food-chain. In addition to TSS, mastitis, and food intoxication SEC seems to play a critical role in the development of infective endocarditis and atopic dermatitis [36,37]. Other diseases that are associated with SEs in general include severe nasal polyposis, perineal erythema, desquamative inflammatory vaginitis, and sudden infant death syndrome [38].

An overview of different SEC variants and their relationships is given in Table 1 and Figure 1. While other *Staphylococcus* species can produce SEC and variants such as SEC_canine_ [39], this review will focus exclusively on *S. aureus* SECs. The great variety of SEs and their genomic location further complicates the intricately-intertwined regulatory pathways of *S. aureus* virulence factors. Previously, SEs in general [4,16,40,41], as well as SEB specifically and its role as a potential bioweapon [42,43] have been reviewed in depth. Information on SEC and its pathogenic role in food intoxication and infection is, however, limited. Potential differences in virulence and toxicity for SEC variants further obscure research efforts. Here, we provide an overview of the role of SEC in foodborne intoxications and clinical manifestations and summarize recent findings in SEC characterization with special regard to the many facets of its variants.

## 2. SEC in Food Intoxication

A striking feature of SEs is their emetic activity. It provokes vomiting, the key symptom in SFP. Some SEs, but not SEC also cause diarrhea [95]. The exact biological purpose of these properties is still unclear. A function in pathogen spread through emesis or diarrhea seems unlikely since *S. aureus* survives very poorly in gastric juice [96]. Still, it has been suggested that *S. aureus* intestinal carriage rate ranges from 8–31% in the healthy population [97]. It is therefore possible that the gastric activity of enterotoxins supports persistence or epithelial barrier invasion of *S. aureus*.

Most SFP outbreaks are attributed to SEA, although there might be some bias since enterotoxins other than SEA–SEE cannot be detected using commercial kits. Many strains produce multiple toxins and it is often unclear which enterotoxin was the cause for SFP. Synergistic effects of different toxins are also likely. Some outbreaks could be traced back to SEC presumably being the only SE involved [4,98,99,100]. SFP can occur with any food that provides sufficient carbon and amino acid sources for *S. aureus* growth. The pathogen is usually introduced by food handlers and produces SEs if conditions allow bacterial growth in the food matrix [12]. Most cases of SFP can be prevented by adequate hygiene measures and intact cooling chains [101]. Due to the heat tolerance of SEs, reheating foods may eliminate *S. aureus* but SEs remain emetically active [3]. SEs are generally very resilient towards external stressors such as heat, acidity, and gastric enzymes [16]. Once SEs reach the small intestine, they enter the lamina propria through mucus-producing goblet cells or epithelial cells [102]. This process may be facilitated in the presence of other *S. aureus* virulence factors [103]. SEA was shown to stimulate 5-hydroxytryptamine (serotonin) and histamine release from mast cells [104,105]. However, it is generally assumed that this principle applies for all emetic SEs in possession of the disulphide loop (see chapter 4). Serotonin acts on the vagus nerve by evoking an emetic response [106]. The role of T-cell and neutrophil activation is unclear, but may be a contributing factor in gut epithelial invasion of *S. aureus* [45]. The involvement of the vagus nerve was demonstrated in very early experiments showing that monkeys did not present emetic symptoms upon SE ingestion after a vagotomy [107]. The basic principle of SE-induced emesis is illustrated in Figure 2.

Whether SEC variants have a different emetic potential is unclear. SEC_1_ [108], SEC_2_ [21], and SEC_3_ [60] have been tested in the monkey-feeding assay, but administered amounts varied, and different extraction methods were applied. SEC_2_ was also tested in the house musk shrew [27]. Therefore, the results cannot be directly compared.

### SEC in Milk and Dairy Products

Intoxications caused by SEs are the most common cause of food poisoning after consumption of raw milk or products made thereof. SEC and SEA are the most-commonly-occurring SEs in milk and dairy products [13,109,110,111,112]. In raw milk samples from Sweden, *S. aureus* was found with a prevalence of up to 71% [113]. This likely stems from the frequent occurrence of SEC in intramammary infections in milk-producing animals [31].

The effect of milk on SE production has been investigated experimentally. When *S. aureus* was grown in milk the expression of SEC was significantly reduced. Downregulation of the *agr* system (see chapter 5) likely contributed to the observed reduction, but other factors are also expected to be involved [114]. The effect of heat treatment on the activity of staphylococcal enterotoxins of type A, B, and C in milk was investigated for pasteurization temperatures [115] and higher temperatures [116]. The amount of detectable SEC could be reduced by heat treatment at 100 °C and above. Another study investigated the production of SEC_bovine_ in milk and during cheesemaking. *S. aureus* numbers increased during cheesemaking but did generally not reach the >10^8^ CFU/mL required for SEC_bovine_ detection in this food matrix. The influence of the added starter culture was not investigated in detail [117]. Recently, the impact of *Weissella paramesenteroides* GIR16L4 or *Lactobacillus rhamnosus* D1 or both together used as starter cultures on the expression of *S. aureus* SEC was examined. The starter cultures were not able to reduce *S. aureus* growth, but they influenced toxin expression in some strains [118]. Lactic acid bacteria or their metabolites could impact quorum sensing of *S. aureus* and therefore influence *agr*-regulated SEs. In another trial, *Staphylococcus vitulinus* was used as a starter culture to successfully inhibit SEC-producing *S. aureus* growth in a barbeque cheese production facility [119].

## 3. Superantigenic Activity of SEC

All SEs are classified as SAgs. In contrast to normal antigens, they bind to MHCII in a location adjacent to the peptide groove. They thus stimulate T-cell receptors (TCRs) in a non-specific way by cross-bridging them with major histocompatibility complex class II (MHCII) on antigen-presenting cells (APC). The resulting global overstimulation of T-cells interferes with immune system functions that normally counteract bacterial infections [38]. In infectious diseases SAgs therefore contribute substantially to transcytosis and immune system evasion [120]. In addition, SEC activation of T-lymphocytes has been exploited for anti-cancer drug development [121]. In spite of advances in SEC research and related drug development, the role of superantigenic activity in food intoxications remains unknown.

The exact location of MHCII binding varies, depending on the SE. Group II antigens, including SEC, bind to the low-affinity α-chain [94]. The resulting mobilization efficiency is 10–100-fold lower for these SAgs than for group III or V SAgs. This is compensated for by the considerably-higher production of these compounds by *S. aureus* [38].

TCR binding is specific to the variable region Vβ of the receptor [17,122] (Figure 3). This results in massive Vβ-dependent T-cell proliferation and subsequent release of pro-inflammatory cytokines [38,94]. Which Vβ subfamilies are stimulated, again depends on the SAg and can even vary between different variants [92]. The following human Vβ-specificity was found for variants SEC_1–3_: C_1_: 3.2, 6.4, 6.9, 12, 15.1; C_2_: 12, 13, 14, 15, 17, 20; and C_3_: 5.1, 12 [123,124,125]. SEC_bovine_ has been shown to specifically activate the Vβ repertoire of cattle [126]. Furthermore, when different SEC variants were compared, SEC_ovine_ strains showed a particularly-strong response in cattle peripheral blood mononuclear cells (PBMCs) [30]. This adaptation of SAgs to mobilize specific T-cell populations contributes to the capacity of *S. aureus* to adapt to different host species [92]. Even in variants not linked to different hosts, functional differences are apparent. When compared in an MHCII-deficient cell line, SEC_1_ was able to still induce T-cell proliferation, but SEC_2_ and SEC_3_ were not. A few specific amino acids near the NH_2_-terminus seem to be responsible for this difference [127]. However, many of the specific differences between variants with regard to the interferon response are not fully understood.

Another factor adding to the immunostimulatory properties of SEs is their ability to co-bind CD28 on T-cells [128]. This has only been demonstrated for SEB [129] but the structural resemblance with SEC suggests that these findings may also apply to SEC.

One of the consequences of the overshooting T-cell proliferation is potentially fatal toxic shock syndrome (TSS). TSS was originally attributed to toxic shock syndrome toxin (TSST-1) and the use of high-absorbing tampons [130]. However, it was demonstrated that non-menstrual TSS can also be caused by SEC and other SEs [81,131]. Long-term exposure to SEs and their superantigenic activity has reportedly been associated with auto-immune diseases such as psoriasis, atopic dermatitis, systemic lupus erythematosus, and, potentially, Kawasaki disease [94]. Food intoxication symptoms like fever and malaise can be attributed to superantigenic activity.

## 4. Physical and Chemical SEC Protein Properties

All SEs share almost identical structural features, although their amino acid sequences vary. In addition, they are structurally related to streptococcal SAgs such as streptococcal pyrogenic exotoxins (SPEs). They are water-soluble peptides with a length of 220–270 aa and have a molecular weight of ~22–29 kDa [4]. Mature SEC, specifically, is 239 aa in length (not including the 27 aa signal peptide at the N-terminus) and has a molecular weight of 27.5–27.6 kDa, depending on the variant.

Sequence similarity of SEs ranges from ~20% to >95%, whereby 15% of amino acids mostly located on the central and C-terminal portions are entirely conserved [16,40]. In regard to its sequence, SEC is most similar to SEB [3]. Within SEC, variants SEC_2_ and SEC_4_ are almost identical with only two aa differences, while SEC_3_ and SEC_ovine_ differ the most with 18 aa. Sequence similarity amongst SEC variants is at least 93% (Figure 1, Appendix A). Interestingly, the signal peptide of SEC_3_ is identical to the one of SEB and therefore differs substantially from the other SEC variants. However, to date there is no indication of a different secretion mechanism [132,133,134,135].

As for many other SEs, the crystal structure of some SEC variants has been resolved [136,137,138,139]. The peptide consists of two unequal domains. The larger one contains a β-grasp fold formed by five β-sheets and an α-helix (β6–β12 & α5). The smaller one comprises a Greek key motif of β-sheets, also known as the oligosaccharide/oligonucleotide fold (OB-fold) found in numerous other bacterial toxins and small α-helixes (β1–β5 and α3, α4, and α6) [4]. The binding sites involved in superantigen activity of SEs have been partially identified [140,141]. The shallow cavity between the two domains binds TCR while major histocompatibility complex class II (MHCII) binding is guided by a region in the N-terminal domain (Figure 1) [4]. It is believed that subtype-specific antigenic epitopes are determined by the N-terminus of SECs, while conserved C-terminal regions define antigenic epitopes shared with other pyrogenic toxins [59].

Like some other SEs and SPEs, SECs have two zinc-binding domains close to the MHCII binding site possibly involved in dimerization and MHCII binding mode [137,138,139]. However, SEC can bind to MHC class II molecules outside the groove on the flanking helix from the α chain via the zinc-independent MHCII binding site [16,138,142].

A distinct disulphide loop (also termed cystine loop) present in all emetic SEs and absent in some weakly- or non-emetic types has been implicated in the toxin’s emetic activity (in the N-terminus between β4 and β5). Structural analyses in SEC have confirmed the important role of the disulphide loop in emesis. Substitution of the two cysteines to alanine in SEC_1_ resulted in loss of emetic activity, however changing the residues to serine did not [143]. This was explained by the ability of serine hydrogen bonds to stabilize the critical loop structure as in the case of the disulphide bond [143]. These findings highlight the importance of the structural conformation, while the chemical makeup itself seems not to be crucial. Therefore, it is the tertiary structure rather than the disulphide bond itself that is responsible for emesis induction.

Mutational analysis in SEA revealed that emetic and superantigenic properties can be attributed to different regions of the peptide, hence separating emesis and superantigenic activity as two different functions of SEs [144,145]. In the case of SEC, it was shown that a lack of the disulphide bond did not affect superantigen activity in an early experiment [143]. However, some articles suggest that even though emetic and superantigen properties are localized in two different regions, they still partly correlate with each other [145]. In particular, superantigen activity seems to depend on multiple domains whereas emetic activity can be allocated more clearly to a specific region [13]. A later study found that regions 21–51 and 81–100 were essential for both emesis and superantigenicity in SEA [146]. To what extent these finding apply to SEC is unclear.

## 5. Genetic Localization and Regulation of SEC

Genes coding for SEC (*sec*) are generally located on a *S. aureus* pathogenicity islands (SaPIs), but plasmids carrying *sec* have also been described [30,55,57]. SaPIs are mobile genetic elements that harbor many virulence factors and likely contribute to *S. aureus* evolution via horizontal gene transfer. They derived from prophages and require helper phages for transmission [93]. The *sec* gene has been described in SaPIbov1, SaPIn1/m1, SaPImw2, SaPIov1, and SePI1. Moreover a BLAST search resulted in SaPINuSaα2, SaPITokyo, and SaPIbov5 (Table 1) [56]. Incidentally, NCBI GenBank contains multiple WGS that include *sec* in regions containing SaPI-related features although they may not be annotated as such. It is often co-localized with *tst-1*, the gene coding for toxic shock syndrome toxin-1 (TSST-1) [57].

The expression of SEC is highest at late exponential to stationary phase [147]. It is regulated by the quorum-sensing system of the accessory gene regulator (*agr*) that uses autoinducing peptides (AIP). It acts on toxin transcription indirectly via *RNAIII* that represses the repressor of toxins (*rot*) [148,149,150,151,152]. Additionally, SarA, σ^B^, and SaeRS may play a role in regulation of SEC when environmental factors change (Figure 4) [45,153,154]. Food-related stressors such as glucose or NaCl have been shown to influence SEC expression. In an early study, glucose led to reduced extracellular SEC concentrations. Furthermore, an intact *agr* system was not required for the effect of glucose on *sec* expression [149]. In a later study, the expression of *sec* was shown to be regulated in response to high NaCl concentrations. The results demonstrated that osmoregulation of SEC occurs at the level of mRNA independently of an intact *agr* allele. Osmoprotective compounds did enhance SEC expression [148]. In an older study on the effect of NaCl and pH on SEC production it was not possible to demonstrate SEC production in broths with 12% NaCl and a pH range of 4.50 to 8.55 [155].

In conclusion, under both glucose and NaCl stress *sec* mRNA and SEC protein levels were considerably reduced [148,149,155]. It is unclear whether the different genetic locations of SEC variants influence transcriptional regulation. Many external stressors and their influence on SEC expression have not yet been investigated.

## 6. Challenges and Future Prospects

Many challenges arise from the great variety of SEs. Firstly, new SEs cannot yet be detected by commercial assays. Therefore, reports on the frequency of SEs associated with intoxication and disease are likely biased towards classical SEs. Secondly, the influence of the food matrix and host factors can influence toxin expression in strain-specific, toxin-specific, and even variant-specific ways. Additionally, the genetic location of SEs may play a role in genetic regulation. Nonetheless, many studies do not account for these aspects. Legislative documents limiting *S. aureus* occurrence by CFU/gram food, regardless of the respective toxin expression, reflect these shortcomings.

For SEC variants in particular, it remains unknown whether genetic location or external stressors result in variant-specific responses in genetic regulation. Furthermore, although agr undoubtedly plays an important role in SEC regulation, additional regulatory elements should be considered in future studies. Whether SEC variants differ in their effects on human and animal health has not been resolved. The different Vβ-specificities of SEC variants likely trigger slightly different superantigenic responses. Hence, some variants might be more relevant in acute diseases while others could be implicated with chronic or persistent progressions. Additionally, variants have never been compared directly with respect to their emetic activity. Therefore, correlations between variants and emetic dosage cannot be provided.

In conclusion, aspects such as toxin variants, strain-specific responses, and influences of external stressors should be considered when investigating SEC expression in the future, and results should not be generalized. Taking all of these factors into consideration will contribute to predicting and maintaining high food safety standards and improving human and animal health.

## Figures and Tables

**Figure 1 toxins-12-00584-f001:**
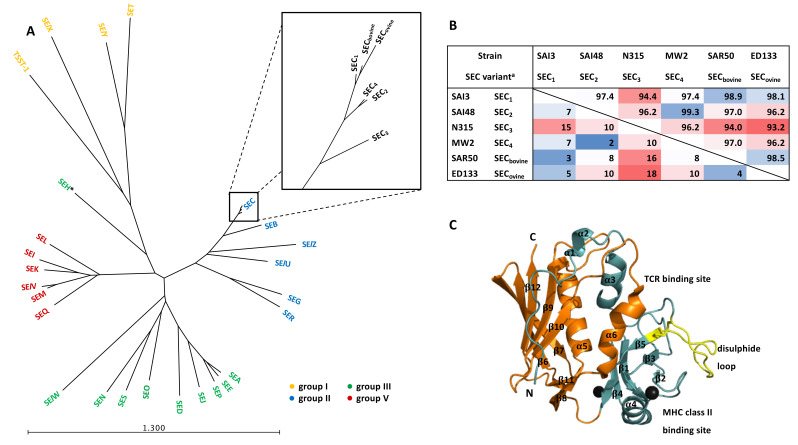
(**A**) Similarity tree based on staphylococcal enterotoxins (SE) sequences including signal peptides—close-up of SEC sequences. The tree was constructed with CLC genomics workbench 12. Colors indicate phylogenetic pyrogenic toxin superantigen (Sag) groups according to Wilson et al., 2018 [92]. * SEH has been assigned to group III or separately in group IV. (**B**) Similarities between SEC variants including signal peptides. The upper right half shows similarity in %. The lower left half shows amino acid differences in absolute numbers. Colors indicate sequence similarity with red = low and blue = high. ^a^ Protein sequences are available under the accession numbers indicated in Appendix A. (**C**) Ribbon diagram of SEC_2_. The N terminal domain is colored in light blue, the C-terminal domain in orange. The disulphide bridge and loop are colored in yellow. Binding sites are labelled. The two zinc ions are indicated with black spheres. α-helices and β-sheets are numbered from N to C terminus. PDB file accession number is indicated in Appendix A.

**Figure 2 toxins-12-00584-f002:**
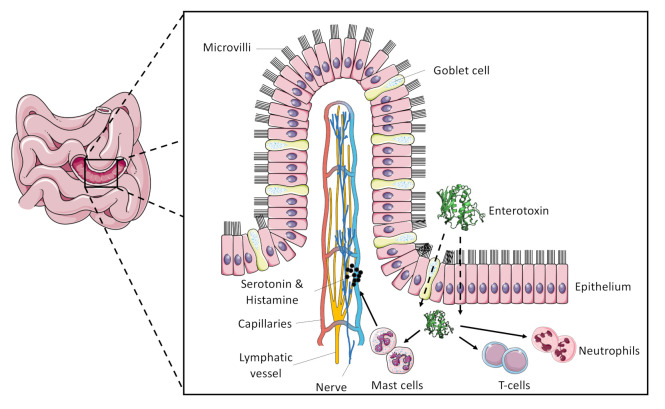
Mechanism of SE emetic activity as proposed by Fisher et al., 2018 [45]. Enterotoxins enter through epithelial or mucus producing goblet cells. The activation of mast cells leads to release of serotonin. Serotonin stimulation of the vagus nerve provokes an emetic response. T-cells and neutrophils are activated as well but their role remains unclear. The figure was adapted from the original publication. Schematic illustrations were created using Servier medical art: https://smart.servier.com.

**Figure 3 toxins-12-00584-f003:**
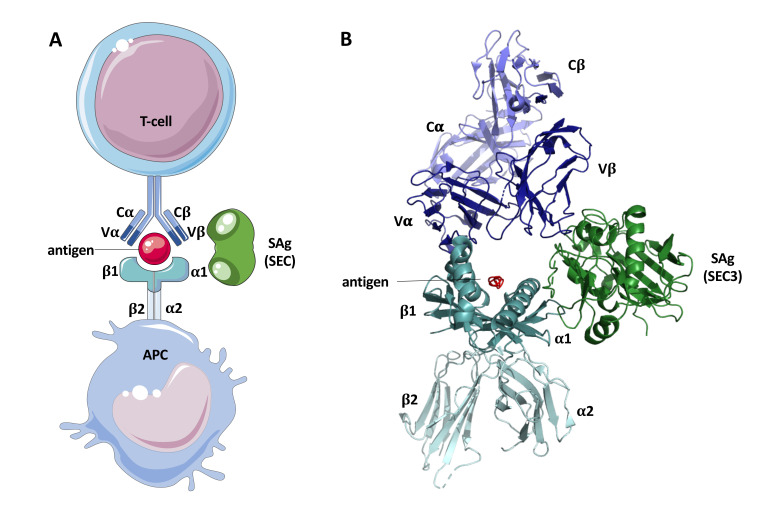
(**A**) Schematic representation of superantigenic activity of SEC. Schematic illustrations are from Servier medical art: https://smart.servier.com. (**B**) Model of SEC_3_ bound to MHC II α1 and TCR Vβ. SEC_3_ is colored in green, TCR in blue, MHC II in cyan, and the normal antigen in red. The model was composed with pyMOL v2.4.0 from two separate models of SEC_3_ complexed to MHC II, and SEC_3_ complexed to TCR. PDB accession numbers can be found in Appendix A.

**Figure 4 toxins-12-00584-f004:**
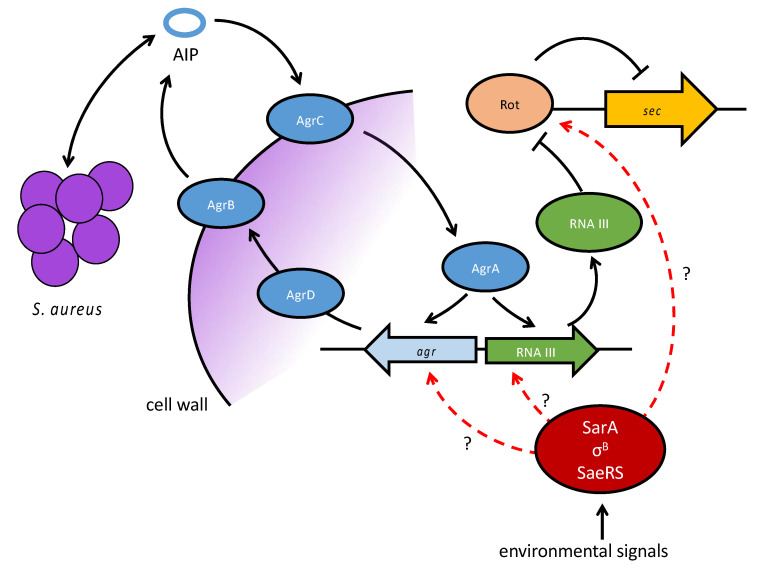
Regulatory pathways involved in sec transcription. The quorum-sensing Agr system acts on sec transcription indirectly. AgrA induces RNAIII which represses the repressor of toxins (Rot), consequently allowing transcription of the sec gene. SarA, σ^B^, and SaeRS might play an additional role in transcriptional regulation when environmental stress signals act on the cells.

**Table 1 toxins-12-00584-t001:** Staphylococcal enterotoxin gene location and emetic activity. Adapted from [4,12,44,45]. nd = not determined.

Enterotoxin/Enterotoxin-Like SAg	Emetic Activity	Associated Genetic Element	References
SEA	+	Prophage (φSa3ms, φSa3mw, φ252B, φNM3, φMu50a)	[19,46,47,48,49]
SEB	+	SaPIs (SaPI1, SaPI2, SaPI3, SaPI4, SaPImw2, SaPIrki4) Plasmid (pZA10)	[50,51,52,53,54]
SEC	+	SaPIs Plasmid	[55,56,57]
SEC_1_	+	SaPINuSAα2 ^a^, pZA10	[53]
SEC_2_	+	SaPITokyo ^a^	[58]
SEC_3_	+	SaPIn1/SaPIm1 ^b^	[59,60]
SEC_4_	nd	SaPImw2	[61]
SEC_bovine_	nd	SaPIbov1	[30,62]
SEC_ovine_	nd	SaPIbov5 ^a^, SaPIov1	[30,62,63]
SED	+	Plasmid (pIB485-like)	[64,65,66]
SEE	+	Prophage(hypothetical)	[67,68]
SEG	+	*egc* (*egc*1–4) Prophage (φSa3ms)	[61,69,70,71,72]
SEH	+	Transposon (MGEwm2/mssa476 seh/Δseo)	[61,73,74,75]
SEI	+	*egc* (*egc*1–3)	[69,70,76]
SE*l*J	nd	Plasmid (pIB485-like, pF5)	[27,77,78]
SEK	+	SaPIs (SaPIbov1, SaPI1, SaPI3, SaPI5) Prophage (φSa3ms, φSa3mw)	[24,61,71,79,80]
SEL	+	SaPIs (SaPIbov1, SaPI3, SaPIn1, SaPIm1, SaPImw2)	[24,57,80,81,82]
SEM	+	*egc* (*egc*1–2)	[24,69,70,80]
SEN	+	*egc* (*egc*1–4)	[24,69,70,80]
SEO	+	*egc* (*egc*1–4) Transposon (MGEwm2/mssa476 *seh*/Δ*seo*)	[24,69,70,80]
SEP	+	Prophage (φSa3n, φN315, φMu3A)	[24,83]
SEQ	+	SaPIs (SaPI1, SaPI3, SaPI5) Prophage (φSa3ms, φSa3mw)	[24,84,85]
SER	+	Plasmid (pIB485-like, pF5)	[77,86]
SES	+	Plasmid (pF5)	[86]
SET	+	Plasmid (pF5)	[86]
SE*l*U	nd	*egc* (*egc*2–3)	[87,88]
SE*l*U_2_ ^c^	nd	*egc* (*egc*4)	[88]
SE*l*V	nd	*egc* (*egc*4)	[88]
SE*l*W	nd	Chromosome	[89]
SE*l*X	nd	Chromosome	[90]
SE*l*Y	nd	Chromosome	[91]
SE*l*Z	nd	Chromosome	[92]

^a^*S. aureus* pathogenicity islands (SaPIs) were determined via NCBI nucleotide BLAST, accession numbers can be found in Appendix A. ^b^ SaPIm1 (MU50) and SaPIn1 (N315) are identical [93]. ^c^ SE*l*U_2_ was suggested to be renamed into SE*l*W but since sequence similarity with SE*l*U is 94% we recommend using SE*l*U_2_ [94], this also facilitates discrimination from chromosomal SE*l*W [89].

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
