# Peer review of "Staphylococcal Enterotoxin C—An Update on SEC Variants, Their Structure and Properties, and Their Role in Foodborne Intoxications"

_toxins, 2020, doi:10.3390/toxins12090584_

Round 1
Reviewer 1 Report
The review article entitled"Staphylococcal enterotoxin C – an update on SEC variants, structure and properties, and their role in foodborne intoxications" is relevant and attracts a wide range of readers. However, the excessive use of short forms in the text may distract the reader.
Some general comments which may be useful to improve the the article.
- Rephrase this sentence Line no 23. "The most common causative agents for food intoxications are staphylococcal 23 enterotoxins (SEs) that are exotoxins preformed" by adding "One of the most"
- Line 30 "Due to the generally" please remove "the"
- Line 31. please add references explaining various diseases caused by Staphylococcal toxin!
- Merge the paragraph line 29-31 with the next paragraph. 2 lines paragraph is inadequate.
- Line 43, please define "SEA-SEE" while using first time.
- Please confirm this "(where < 10 % sequence divergence equals a variant)" in line 52. It should be more than 10%?
- Please this unnecessary sentence "please refer to the cited literature." Line no 67
- In the figure legend why the sentence explaining the figure is half bold and half normal font. It is present in all figures. Please correct if this is not the journal format.
-
What are these 8-31% SA strains, Are the acid resistant? please name some in the sentence.
-
Please elaborate: Gastric activity of enterotoxin supports persistence? What is the proposed or known mechanism?
- Please explain "SE being heat tolerant how can be these be inactivated still retain the emetic activity?" Line number 99.
- Line 230. via homologous gene transfer? please replace it with horizontal gene transfer.
Reviewer 2 Report
Manuscript ID: toxins-905136
This review describes the SEC variants, structure and properties, and their role in foodborne intoxications. This review provides recent findings on SEC and novel perspectives in food safety. Your kind consideration of the following points would be sincerely appreciated.
Lines 58-59:
Many of the methicillin-resistant Staphylococcus aureus (MRSA) involved in major nosocomial infections have been reported to be SEC producing strains. Since many SEC-producing strains are in MRSA, is SEC involved in human post-partum mastitis? Or is SEC -producing strains specifically involved in human post-partum mastitis in the presence of various SE-producing strains?
Lines 59-61:
Is SEC specifically involved in infective endocarditis, atopic dermatitis, severe nasal polyposis, perineal erythema, desquamative inflammatory vaginitis, and sudden infant death syndrome? Are other SEs associated with these diseases similarly?
Table 1:
Are references of SEC1~SECovine the same as references of SEC?
Figure 1C:
The meaning of numbers and symbols needs to be added to Figure legend.
Lines 92-93:
If possible, please describe the contribution ratio of SEC to SFP and the presence ratio of SEC-producing strains in the food environment.
Lines 103-104:
It was reported that SEA induces histamine release from submucosal mast cells in the gastrointestinal tract. Histamine contributes to the SEA-induced vomiting reflex via the serotonergic nerve (and/or another vagus nerve).
Please refer to the following reference.
Ono, H. K. et al., Histamine release from intestinal mast cells induced by staphylococcal enterotoxin A (SEA) evokes vomiting reflex in common marmoset. PLoS pathogens, 15(5): e1007803 (2019).
Lines 125-126:
What percentage of the isolated S. aureus strains had the SEC gene?
Is SEC-producing S. aureus strain of animal origin transmitted to humans via food?
Lines 156-157:
How much toxin production is in Group II (SEC) compared to Group I?
Is there a difference in SEC production due to the difference in SaPI type?
Lines 223-225:
Emetic and superantigenic activities have also been reported to be activated by the same region of the SEA peptide. (Amino acid residues 35–50 and 81–100 of SEA molecule are necessary to superantigenic and emetic activities. (Maina, E. K. et al., 2012))
Are the emetic activity of SEC and other SEs need for the superantigenic properties?
Maina, E. K. et al., Inhibition of emetic and superantigenic activities of staphylococcal enterotoxin A by synthetic peptides. Peptides, 38(1): 1-7 (2012).
Line 263:
The scientific name should be italic.
